# Ageism, Job Engagement, Negative Stereotypes, Intergenerational Climate, and Life Satisfaction among Middle-Aged and Older Employees in a University Setting

**DOI:** 10.3390/ijerph19137554

**Published:** 2022-06-21

**Authors:** Jasmin T. McConatha, V. K. Kumar, Jaqueline Magnarelli

**Affiliations:** Department of Psychology, West Chester University of Pennsylvania, West Chester, PA 19383, USA; kkumar@wcupa.edu (V.K.K.); jm921309@wcupa.edu (J.M.)

**Keywords:** ageism, life satisfaction, stereotypes, discrimination, work engagement

## Abstract

This study examined whether age-related discrimination, negative age-related stereotypes about declining abilities due to age, job engagement (cognitive, physical, and emotional), and workplace intergenerational climate in terms of positive intergenerational affect (PIA) and workplace intergenerational inclusiveness (WIG) correlated with life satisfaction in a university setting. The analysis was based on 115–117 faculty and staff, 50 years or older. A Principal Axis factor analysis with Promax rotation on the job-related variable revealed three factors: Experiencing Ageism (discrimination and negative stereotypes), Work Climate (PIA and WIG), and Job Engagement (physical, emotional, and cognitive). The factor-based regression scores on the three-factor-based scores were correlated with life satisfaction and also subjected to hierarchical regression analyses with age, sex, and education entered on the first step and the three factors on the second step. The results of both the correlational and hierarchical regression analysis indicated that experiencing ageism was significantly predictive of life satisfaction, and that ageism may play a more primary role than job engagement and work climate-related variables in accounting for life satisfaction.

## 1. Introduction

Life satisfaction, the perceptions of one’s quality of life, is an important psychological variable shaping overall wellbeing. Important variables in work environments that may be related to life satisfaction are experiencing age-based discrimination, perceiving discrimination as prevalent (how often one receives maltreatment or disrespect [1]), knowing that popular age-related negative stereotypes exist in the workplace, and an intergenerational work climate, as well as job engagement. In the university setting, life satisfaction has been related to job satisfaction, mobbing, time pressures, academic performance, and relative income [2]. 

Ageism in the academic setting is both overt and covert [3]. For example, studies of ageism in academia have found that younger colleagues may patronize older faculty, particularly women [4]. Older women faculty are also more likely to be excluded or subjected to “mom-ism,” while younger women faculty are more likely to be viewed as more committed, productive, and energetic [5,6,7]. However, studies that have examined life satisfaction and ageism-related variables in a university setting are rare. 

Discrimination against older workers, like other forms of discrimination, is a serious social and human rights concern. Ageism, a widespread form of prejudice, is a devastating social phenomenon that has not been widely acknowledged and researched. Ageism is defined as holding stereotypical, prejudicial, and discriminatory attitudes against individuals or groups simply because of their age [8]. Unlike racism and sexism, ageism is often seen as “normal”, particularly in the workplace [8], something that is taken for granted. It is a form of discrimination not often discussed in the workplace or elsewhere. As people live longer, healthier lives, they also remain in the workplace longer [9]; however, a majority (64%) of older workers, particularly female workers, report experiencing or witnessing age discrimination in the workplace [10,11], with some studies indicating that over 80% of women over 50 report having experienced age-related discrimination [3].

The number of Age Discrimination in Employment Act (ADEA) complaints filed by those older than age 55 has steadily increased since 2000, during which their numbers have grown in the workplace [12,13]. Older workers report feeling that their contributions are unacknowledged, and they feel left out of decision-making and planning functions [14]. Ageist attitudes result in fewer work opportunities, training, promotions, and retention [15,16,17]. An AARP [15] survey revealed that older respondents who reported age-related discrimination reported having experienced negative remarks concerning their older age from a colleague or supervisor, not being hired for a job, being passed over for a promotion or another advancement opportunity, and being laid off, fired, or forced out of a job. This especially seems to be the case for women. The Bureau of Labor Statistics [18] predicted that by 2024, there will be twice as many women workers over the age of 55 in the labor force as those women between the ages of 16 and 24. In addition, as the number of older women in the workforce has increased, the number of age discrimination complaints filed has increased. In 1990, men filed nearly twice as many ADEA complaints as women; however, in 2010, more women filed complaints about experiencing age bias than men for the first time [12]. This trend seems to be continuing, with more women filing age discrimination complaints each year. Moreover, most reports of age bias in 1990 were from workers between the ages of 40 and 54, but in 2017, workers between the ages of 55 and 64 filed the most age discrimination complaints. The percentage of charges filed by workers aged 65 and older in 2017 was twice what it was in 1990 [11,12]. Often struggling silently, older workers tend not to discuss their mistreatment [19]. An AARP [15] survey revealed that only 3% of respondents who experienced age discrimination made an official complaint—indicating that the problem is significantly more widespread than documented.

Ageist treatment has numerous significant negative social and psychological implications for older workers. It increases stress, negatively impacts self-worth, and can lead to the alienation and isolation of older workers. Fernandez-Ballesteros et al. [20], reviewing the literature on the effects of ageism, indicated that older adults tend to suffer many negative (physical, social, mental) outcomes because of ageism—perceiving age-related discrimination and negative age-related stereotypes and prejudices. In their large sample study in three countries: Mexico, Germany, and Spain, they found that perceived discrimination correlated significantly with life satisfaction in Spain (*r* = −0.324, *p* < 0.01) and Mexico (*r* = −0.319, *p* < 0.01), but not in Germany (*r* = −0.091).

Firzly et al. [21] observed that workplaces free of age-based stereotypes are associated with higher levels of worker satisfaction and decreased intention to retire. The perception that coworkers and colleagues hold negative attitudes about older workers may have important consequences for both younger and older workers and the organization. Perceived ageism, discrimination, and age-based stereotypes are associated with reduced self-esteem, employment opportunities because of biases in hiring decisions [22], job satisfaction [23,24], and work engagement [25]. 

The results of a meta-analysis [26] indicate that age-based negative stereotype threats about competence held by older workers can negatively impact their memory (*d* = 0.21) and cognitive performance (*d* = 0.68), and these effects remain over different sex and age groups. Furthermore, there is evidence for greater vulnerability when induced by stereotype threats (*d* = 0.52) than when induced by facts (*d* = 0.09). Jang et al. [1] argued that perceived discrimination is unpleasant and stressful and leads to reduced feelings of wellbeing. Using a large sample of 45–74-year-old adults, they found evidence for not only the direct effects of perceived discrimination on feelings of wellbeing, but also for indirect effects through reducing a sense of control for both positive and negative affect. Yao et al. [27] found that perceived discrimination has a direct negative impact on the life satisfaction of older adults and an indirect impact that occurs through identity and community sense. Redman and Snape [28] found, in a sample of police officers, that perceived age discrimination was negatively associated with their job and life satisfaction, power and prestige associated with their jobs, and commitment. In their study, job involvement did not have a significant correlation with life satisfaction. They concluded that discrimination is a significant stressor with severe psychological consequences.

Levy [29,30] formulated the Stereotype Embodiment Theory (SET), which holds that age-based discrimination, popular negative stereotypes about older people, and negative self-perceptions held by older people about their aging can have serious effects on the health and wellbeing of older individuals. Levy et al. [30] noted that there has been extensive research across five continents that yielded findings consistent with predictions from the SET theory that ageism negatively impacts the health of older individuals. Levy et al. [30] found that the health care costs associated with negative self-perceptions of aging were much higher than those associated with negative age stereotypes, followed by those associated with age discrimination; adjusted for age and sex, the “excess cost was $11.1 billion for age discrimination, $28.5 billion for negative age stereotypes, and $33.7 billion for negative self-perceptions of aging” [30] (p. 178).

A variable of interest to the present study was intergenerational contact. Hanrahan et al. [22] concluded, from a review of studies (e.g., [23]), that positive intergenerational contact can help ameliorate the negative impact of ageism, which in turn can promote intergroup harmony among older and younger workers. King and Bryant [31] found a correlation (*r* = 0.58) between intergenerational climate at work and job satisfaction in their Study 2. A recent Canadian study by Firzly et al. [21] found, contrary to their expectations, that perceiving that the workplace intergenerational climate was positive was associated with decreased awareness of ageist practices; they explained this unexpected finding by suggesting that if things are going well for them, then it is also good for the older workers. However, consistent with their expectations, the intergenerational climate was associated with increased job satisfaction, and sharing and donating knowledge behaviors. Such behaviors were, in turn, associated with greater awareness of ageism against older workers and greater job satisfaction. 

Job engagement or work engagement has also been explored as a work-related motivational construct. Rich et al. [32] developed an instrument to measure the three dimensions proposed by Kahn [33]: physical (intensity of effort), cognitive (mindfulness, vigilance, attention to work), and emotional (affect associated with work) energies devoted to their work. They found their scale to be significantly correlated with measures of job involvement, job satisfaction, value congruence (meaningfulness of work), intrinsic motivation, task performance, and other variables. Although Rich et al. [32] did not examine the relationship of their job engagement scale with life satisfaction, there are prior studies that suggest that the two constructs are significantly correlated (e.g., [34,35,36]).

Given that work and the work environment constitute a major portion of one’s life, age-based discrimination, perceiving age discrimination, and knowing that popular negative stereotypes about an age-related decline in abilities exist should have a bearing on how satisfied older people feel about their lives. Few studies seem to have addressed the association of ageism-related variables of discrimination and negative stereotypes with life satisfaction. Thus, the present study was designed to examine whether experiencing age-related discrimination, negative age-related stereotypes regarding declining abilities due to age, job engagement (cognitive, physical, and emotional), and workplace intergenerational climate in terms of positive intergenerational affect and workplace intergenerational inclusiveness correlated with life satisfaction in a university setting. 

Based on the studies discussed previously, it was expected that faculty and staff in middle and later adulthood (age 50 plus) who (a) have experienced lower levels of discrimination are more likely to report experiencing higher levels of life satisfaction, (b) perceive a lower prevalence of negative stereotypes about aging in the workplace are more likely to report higher levels of life satisfaction, (c) perceive higher levels of intergenerational positive affect and inclusiveness are more likely to report higher levels of life satisfaction, and (d) are more physically (effortful working), emotionally (enjoyment), and cognitively engaged in their work are more likely to report higher levels of life satisfaction. On an exploratory basis, the incremental and unique variance accounted for by these selected job-related variables beyond the demographic variables of age, sex, and education through hierarchical regression analysis was also explored.

## 2. Materials and Methods

### 2.1. Participants and Procedures

An email containing a link to an online survey was sent to 1015 instructional faculty members and 805 staff and administrators at a mid-sized university in southeast Pennsylvania. The participants were informed that their participation was voluntary and there was no penalty for not completing the survey, and that their responses were being recorded anonymously. After the participants read the informed consent form approved by the university’s Institutional Board, they were able to continue to the questionnaires, if they consented. 

Although 364 individuals responded to the survey, because of omissions and/or declining to answer some items, the numbers of respondents varied for different questions. Analysis was restricted to 115–117 midlife and older employees (≥50 years of age) who responded to the questionnaire. The respondents ranged in age from 50 to 79 (*M* = 59.57), with 77 females and 40 males, 78 instructional faculty, four administrators, and 35 staff members. Level of Education was coded in five ordinal categories: High School Graduate (including some college, but no degree) (*n* = 9), Associate Degree (*n* = 6), Bachelors (*n* = 15), Master’s (*n* = 23), and Doctoral/Professional (including JD, MD) (*n* = 64). Most respondents identified themselves as White, non-Hispanic (*n* = 93). Few identified as African American/Black (*n* = 5), Asian (*n* = 2), Hispanic or Latino (*n* = 3), More than one ethnicity (*n* = 2), Middle Eastern (*n* = 1), and declined/omitted (*n* = 11)

### 2.2. Instruments

*The Satisfaction with Life Scale (SWLS)* [37] is a five-item scale used to measure one’s satisfaction with life. The scale uses a seven-point Likert scale, with 1 = “*strongly disagree*” and 7 = “*strongly agree”*. Examples of items include “*In most ways, my life is close to my ideal*”, and “*The conditions of my life are excellent*”. The Cronbach α for the scale in the present study was 0.90.

*The Workplace Age Discrimination Scale (WADS)* [38] is a 26-item scale used to assess the prevalence of age discrimination experienced by workers; however, it was modified to a 23-item scale for this study to make the items more relevant to the intended participants (faculty members). The three removed items were not relevant for the university sample in this study because they pertained to promotions, salaries, and responsibilities that are specified by Union Policies. A five-point Likert scale (1 = “*never”*, 5 = “*very often”*) was used to assess discrimination frequency. Higher scores on the scale reflect a higher prevalence of age discrimination experienced by workers. Examples of items include *“I have been passed over for a work role/task due to my age”,* and *“My contributions are not valued as much due to my age”.* The Cronbach α for the scale in the present study was 0.96.

*The Workplace Intergenerational Climate Scale (WICS)* [31] is a 20-item scale to evaluate the relationship between younger and older co-workers. The original scale contained five subscales with four items each: lack of generational stereotypes, positive intergenerational affect, intergenerational contact, workplace generational inclusiveness, and workplace intergenerational retention. However, for this study, only the *Positive Intergenerational Affect* and the *Workplace Generational Inclusiveness* subscales, with four items each, were included. The participants rated each of the items on a four-point Likert scale (1 = “*strongly disagree”,* 4 = *“strongly agree”*). The Positive Intergenerational Affect items include: “*I feel comfortable when co-workers outside my generation try to make conversation with me”, “I enjoy interacting with co-workers that are outside may generation”, “My coworkers outside my generation are interesting and unique individuals”, and “People work best when they work with others of different ages”.* The Workplace Generational Inclusiveness items include: *“I believe that my work environment is a healthy one for all ages”, “Workers of all ages are respected in my workplace”, “I am able to communicate effectively with workers of different generations”, and “Working with co-workers from different generations enhances the quality of my work life”.* Higher scores on the scale indicate more positive intergenerational feelings and perceptions of more inclusiveness of all age groups in the workplace. The Cronbach’s α for the PIA and WGS subscales were 0.78 and 0.80, respectively. The Cronbach’s α for the sum of two scales (Workplace Intergenerational Climate) in the present study was 0.85.

*The Negative Aging Meta-Stereotypes scale* [39] is a seven-item scale designed to measure how people believe other people will stereotype them because of age. However, only four items were used in this study because they were most relevant in a university setting. The items were rated on a five-point Likert scale (1 = “*totally agree*” to 5 = “*totally disagree”*). The scale was reverse-scored so that higher scores reflect higher beliefs in the prevalence of negative aging stereotypes. The items included were “*I believe the majority of my colleagues think that performance declines with age”, “I believe the majority of my colleagues believe that older co-workers resist change”, I believe the majority of my colleagues believe that older co-workers are not interested in learning new skills”, and “I believe the majority of my colleagues feel negative about older workers”.* The Cronbach α for the scale in the present study was 0.85.

*The Job Engagement Scale* [32] is an 18-item scale used to measure three dimensions of job engagement, with six items each: physical, emotional, and cognitive engagement. Items are rated on a seven-point Likert scale (1 = “*Strongly Disagree”* to 7 = “*Strongly Agree”*). The *Physical Engagement* subscale includes such items as “I work with intensity on my job” and “I exert my full effort to my job”. The *Emotional Engagement* subscale includes such items as “I am enthusiastic in my job”, and “I feel energetic at my job”. The *Cognitive Engagement* subscale included such items as “At work, my mind is focused on my job”, and “At work, I pay a lot of attention to my job”. On these scales, higher scores reflect higher engagement. The Cronbach’s α for the three subscales Physical, Emotional, and Cognitive Engagement were 0.93, 93, and 0.94, respectively. The reliability of the Job Engagement total score of the three subscales was 0.96.

*Demographic Variables***.** Participants were also asked to respond to demographic questions about their sex (male, female), job category (staff, administrator, and instructional faculty), age (to the nearest year), education (high school diploma or equivalent, some college but no degree, an associate degree in college—2 years, Bachelor’s degree in college—4 years, Master’s degree, doctoral degree, professional degree (MD, JD)), and ethnicity (White Caucasian—Non Hispanic, African American/Black, Hispanic or Latino, Asian, Middle-Eastern, and more than one ethnicity). All questions allowed the option “decline to answer”.

## 3. Results

### 3.1. Demographics and Life Satisfaction

None of the demographic variables examined (age, sex, education, and employee status) had a significant association with life satisfaction. Age had a correlation of −0.082, *p* = 0.380. The analysis of variance on sex yielded an *F* < 1.0; education, *F* = 1.21, *p* = 0.309; and employee status, *F* = 1.28, *p* = 0.275 in all cases. 

### 3.2. Preliminary Analyses

Given that the sample consisted of 78 instructional faculty, four administrators, and 35 staff members, an initial analysis was conducted to see whether the data could be combined across these groups for various analyses. Additionally, because there were only four administrators, their data were combined with 35 staff members to form one group (administrators and staff). A multivariate analysis on the eight variables (life satisfaction, work discrimination, negative stereotypes, climate—PIA, climate—WGI, physical engagement, emotional engagement, and cognitive engagement) revealed that the mean score for the two groups did not differ significantly (Mult *F*(8, 107) < 1.0) on these variables. A further univariate analysis also showed that the mean scores did not differ significantly on any of the variables and all effect sizes (η^2^) lower than 0.024 for all variables. Thus, the rest of the analyses were performed on all participants.

### 3.3. Correlations among Variables 

Table 1 provides Pearson correlation coefficients between each of the job-related variables and life satisfaction and also among the job-related variables. As expected, (a) experiencing discrimination in the workplace had a negative significant correlation with life satisfaction (*r* = −0.205, *p* < 0.028), indicating that those who reported higher levels of discrimination at work were more likely to report lower levels of life satisfaction, (b) agreeing more strongly that negative stereotypes exist at the workplace correlated significantly and negatively with life satisfaction (*r* = −0.259, *p* < 0.008), indicating that those who reported a lower prevalence of negative stereotypes were more likely to report higher levels of life satisfaction, (c) contrary to expectations, neither of the climate variables nor the total climate score correlated with life satisfaction, and (d) neither physical nor cognitive engagement had significant correlations with life satisfaction, but emotional engagement had significant positive correlation (*r* = 0.249, *p* < 0.008). 

An examination of Table 1 suggests that Work Discrimination and Negative Stereotypes had a significant strong positive correlation (*r* = 0.543, *p* < 0.008). Furthermore, the two climate variables had strong significant positive correlations with each other (*r* = 0.594, *p* < 0.008) and the three job engagement scales had strong significant positive correlations with each other (0.673 to 0.830). 

Given these strong correlations among the predictor scales, we conducted a Principal Axis Factor Analysis (PFA) with Promax rotation (kappa = 4) to allow for correlated factors and to reduce the number of variables to avoid issues of multicollinearity among the predictor variables in a hierarchical multiple regression analysis. The Promax rotation yields two matrices: pattern and structure, and the former matrix is most often interpreted. The PFA yielded three factors, which accounted for 82.420% of the total variance, with eigenvalues of 2.604 (37.196%), 2.086 (29.508%), and 1.100 (15.716%), respectively. The Kaiser–Meyer–Olkin measure of sampling adequacy was 0.63, suggesting adequate sampling. 

Table 2 provides a summary of the PAF analysis rotated pattern matrix. The results suggested three factors to underlie the seven variables: Job Engagement (combination of physical, emotional, and cognitive aspects), Experience of Ageism (combination of work discrimination and negative stereotypes), and Work Intergeneration Climate (combination of the two climate variables). 

Regression-based factor scores were then computed for each of the factors and correlated with Life Satisfaction, which were as follows: Job Engagement *r* = 0.057 (*p* = 0.544), Experience of Ageism = *r* = −0.245 (*p* = 0.008), and Work Intergenerational Climate *r* = 0.070 (*p* = 457), respectively. Factors 1 and 2 correlated = −0.147, *p* = 0.118, factors 1 and 3 correlated = 0.066, *p* = 0.482, and factors 2 and 3 correlated = −0.649, *p* = 0.000, suggesting that higher levels of ageist experiences and attitudes are significantly associated with lower levels of expressed positive intergenerational affect and inclusiveness climate.

### 3.4. Regression Analysis: Predicting Life Satisfaction

A hierarchical regression analysis was conducted to examine the additional variance accounted for in the prediction of Life Satisfaction by the job-related factor variables Ageism, Work Intergenerational Climate, and Job Engagement, beyond that accounted for by the demographic variables of age, sex, and education. The regression analysis permitted examining the relative variance accounted for by each of the job-related variables. There was no evidence of heteroskedasticity in either model; however, as per Hayes and Cai’s [40] recommendation, we employed robust standard errors (HC3) that do not assume heteroskedasticity.

The three demographic (Sex, Age, Education) variables were included in the first step, and three job-related variables (Ageism, Work Intergenerational Climate, and Job Engagement) were included in the second step.

Per Table 3, prediction with demographic variables alone (Model 1) was not significant (*R*^2^ = 0.023, *p* = 0.458); they collectively accounted for about 2.3% of the variance, but when the factor-based variables were added in the second step (Model 2), an additional 6.4% of the variance (*R*^2^ = 0.087, *p* = 0.060) was accounted for. Table 3 shows the results of the regression analysis for predicting life satisfaction in Models 1 and 2.

Per Table 4, in Model 2, only Ageism had a significant beta weight (*b* = −2.386, *t* = −2.545, *p* = 0.0120, suggesting that ageism is a significant predictor of life satisfaction with other variables partialled out; the negative regression coefficient indicates that people who experienced a higher degree of ageism expressed lower levels of life satisfaction. The tolerance values ranged between 0.519 and 0.985, all greater than 0.10, suggesting acceptable levels for the absence of multicollinearity.

## 4. Discussion

Considering the results of the correlational analysis, as expected given prior studies [20,21], individuals who reported more age-related discrimination and age-related meta-stereotypes expressed lower life satisfaction. Moreover, respondents who were more emotionally engaged with their jobs were more likely to express greater life satisfaction. None of the other correlations reached significance. Almost all correlations were small, suggesting that these variables probably play a small role in accounting for life satisfaction. The rather low, but significant, correlation between age-related discrimination and life satisfaction seems surprising given that the work environment tends to be a major part of life, and experiencing discrimination would seem to be associated with life satisfaction to a higher degree. This result can perhaps be explained by the lower variability of the work-discrimination variable. An examination of the frequency distribution suggests our sample reported experiencing rather low levels of discrimination. The scores ranged between 23 and 103, and 23 was the model score with 25.86% (*n* = 30 out of 116) of the participants responding 1 = Never experiencing any discrimination on any of the items; another 26 (22% reported two scores of 24–29), and only 19 respondents reported total scores of 70–103 and above (corresponding to the average rating between Sometimes (3) and Very Often (5)). It is possible that the occurrence of age-related discrimination may be low in this research setting. However, despite guarantees of the anonymity of responses, it is possible that many respondents were not fully open because of the fear that their responses on such sensitive matters can be tracked online. 

Given the significant correlations among the subscales of the included job-related variables, the decision was made to conduct a factor analysis to reduce the number of variables for multiple regression analysis to predict life satisfaction. The factor analysis revealed three factors: Job Engagement (combination of physical, emotional, and cognitive), Experiencing Ageism (combination of age discrimination and stereotypes), and Work Climate (combination of intergenerational affect and inclusiveness). When factor-based correlations for the three factors were correlated with life satisfaction, ageism was the only variable that correlated significantly. The result that climate and job engagement were not predictive of life satisfaction was contrary to our expectation based on earlier studies [21,31,34,35,36]. Furthermore, when life satisfaction was regressed on demographic variables, *R^2^* was not significant, but when the three factors were added in the second step, the change in *R^2^* reached marginal significance. Additionally, controlling for other factors, ageism was the only significant predictor of life satisfaction. The results taken together suggest the relatively greater importance of ageism in the prediction of life satisfaction compared with job engagement and work intergenerational affect and inclusiveness climate. These results are supportive of the results of Yao et al. [27], Redman and Snape [28], and others, but they also add to the previous work that suggests ageism may play a more primary role than job engagement and work climate-related variables. The rather high and significant correlation (*r* = −0.649) between ageism and climate suggests that these factors may go hand in hand, but job engagement may stay independent of both ageism and climate, perhaps because of the high social desirability of the job engagement construct that resulted in a negatively skewed distribution even on the combined variable based on factor analysis.

## 5. Conclusions

Although the respondents reported low levels of workplace discrimination, ageism was significantly predictive of life satisfaction. In addition, the results suggest that ageism may play a more primary role in accounting for life satisfaction than job engagement and climate-related variables in accounting for life satisfaction. These results suggest that institutions should intensify efforts to reduce ageism via policies, procedures, and educational efforts. The present study findings are limited by a large number of individuals choosing not to respond to the survey and, thus, the findings are to be considered preliminary. All of the zero-order correlations are small, even though some of them were found to be significant. The results of the study need to be replicated with a much larger sample through which it will be possible to obtain greater variability on these variables. Given the changing global demographics and the increase in the number of older workers, an exploration of factors that influence satisfaction with work and life are important areas that warrant further investigation. 

## Figures and Tables

**Table 1 ijerph-19-07554-t001:** Correlations among the variables (numbers vary between 115 and 117).

Variable	1	2	3	4	5	6	7	8
1. Life Satisfaction	0.90							
2. Work Discrimination	−0.205 *	0.97						
3. Neg. Stereotypes	−0.259 **	0.543 **	0.86					
4. Climate—PIA	−0.070	−0.141	−0.044	0.80				
5. Climate—WGI	0.121	−0.556 **	−0.365 **	0.594 **	0.82			
6. Physical Engagement	0.036	−0.095	−0.098	−0.009	0.059	0.93		
7. Emotional Engagement	0.249 **	−0.139	−0.129	0.088	0.252 **	0.695 **	0.93	
8. Cognitive Engagement	−0.012	0.020	−0.027	−0.038	0.048	0.830 **	0.673 **	0.95

* *p* < 0.028; ** *p* < 0.008; diagonal values = Cronbach α; PIA = positive intergenerational affect, WGI = workplace generational inclusiveness.

**Table 2 ijerph-19-07554-t002:** Rotated pattern matrix loading of the predictor variables.

Variable	Factor 1	Factor 2	Factor 3
Work Discrimination	0.026	**0.860**	0.040
Negative Stereotypes	−0.016	**0.714**	0.129
Climate—PIA	−0.022	0.231	**0.798**
Climate—WGI	0.036	−0.270	**0.779**
Physical Engagement	**0.919**	−0.036	−0.063
Emotional Engagement	**0.748**	−0.049	0.104
Cognitive Engagement	**0.914**	0.086	−0.036

**Table 3 ijerph-19-07554-t003:** R and R^2^ change from Models 1 and 2.

Model	*R*	*R^2^*	*R^2^* Change	*F* Change	df_1_	df_2_	*p (R^2^* Change)
1	0.152	0.023	0.023	0.872	3	111	0.458
2	0.296	0.087	0.064	2.540	3	108	0.060

Model 1: Demographic Variables: age, sex, Education; Model 2: Job-Related Factor-Based Variables. Adjusted R^2^ for models 1 and 2 were 0.003 and 0.037, respectively.

**Table 4 ijerph-19-07554-t004:** Regression analysis: *t*-test based on robust standard errors (SE).

Variables		*B*	*β*	Robust SE	*t*	*p*
**Model 1**						
Constant	28.265					
Sex *		0.429	0.032	1.240	0.345	0.731
Age		−0.099	−0.094	0.107	−0.926	0.356
Education		0.615	0.121	0.392	1.567	0.120
**Model 2**						
Constant	24.478					
Sex		−0.341	−0.025	1.313	−0.260	0.795
Age		−0.018	−0.017	0.118	−0.150	0.881
Education		0.603	0.118	0.391	1.542	0.126
Job Engagement		−0.039	−0.006	0.590	−0.066	0.947
Ageism		−2.386	−0.338	0.938	−2.545	0.012
Climate		−0.916	−0.135	1.031	−0.888	0.376

* Female = 1, Male = 2.

## Data Availability

Not applicable.

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
