# Peer review of "Ageism, Job Engagement, Negative Stereotypes, Intergenerational Climate, and Life Satisfaction among Middle-Aged and Older Employees in a University Setting"

_ijerph, 2022, doi:10.3390/ijerph19137554_

Round 1

Reviewer 1 Report

Dear editor and authors, thank you for allowing me to review the article "Ageism, Job Engagement, Negative Stereotypes, Intergenerational Climate, and Life Satisfaction Among Middle-aged and Older Employees in a University Setting". This article sought to evaluate the relationship between life satisfaction, ageism and work variables in a university context. The article was made with rigor and quality. Despite not presenting extraordinary results, but considering that it is an exploratory study and in a context where little is known about life satisfaction, this study should be valued. Throughout the review I identified some topics that can be improved, I leave them below, I hope it helps:

On line 26, on the first page, the purpose of the study appears out of nowhere. The objective should come at the end of the introduction. Before presenting the objective, it would be more interesting to show us the rationale of the article, the arguments, which will then justify the objective of the study. At the end of the introduction you present the objectives of the study very clearly, I suggest eliminating this sentence from the first page so as not to become repetitive.

This article innovates in the sense that studies that approach ageism in the university context are very scarce.

The intro is clear and well-organized.

In the participants section, the first paragraph mainly refers to ethical and procedural issues. That's why I suggest that this section be called "Participants and Procedures"

When reading the participants' section, I had a concern that I want to share with the authors. Is there not in your study an important confounding factor associated with sampling (particularly with regard to professional status)?

I don't know if we were able to rigorously put 78 instructional faculty and 35 staff members in the same sample. Do you think ageism and negative stereotypes impact these two subpopulations in the same way? Before proceeding with predictive analyses, I would suggest that you perform some simple comparative analyzes between these populations (e.g., t test, using exact test) in relation to the main variables used in the study (mainly ageism and life satisfaction). Only in this way will we be more confident that all these people can be part of the same group for the analyses.

The instruments are very well described and showed excellent reliability indicators.

Table 1 is unformatted, it is not possible to identify the correlational values correctly

The regression results do not bring much new information considering what has already been shown in the correlational analyses. But they are well presented, organized and thought out.

I found it interesting and enjoyable the way the authors used factor analysis

The discussion adequately discusses the focal results that were found

Author Response

Dear editor and authors, thank you for allowing me to review the article "Ageism, Job Engagement, Negative Stereotypes, Intergenerational Climate, and Life Satisfaction Among Middle-aged and Older Employees in a University Setting". This article sought to evaluate the relationship between life satisfaction, ageism and work variables in a university context. The article was made with rigor and quality. Despite not presenting extraordinary results, but considering that it is an exploratory study and in a context where little is known about life satisfaction, this study should be valued. Throughout the review I identified some topics that can be improved, I leave them below, I hope it helps:

On line 26, on the first page, the purpose of the study appears out of nowhere. The objective should come at the end of the introduction. Before presenting the objective, it would be more interesting to show us the rationale of the article, the arguments, which will then justify the objective of the study. At the end of the introduction you present the objectives of the study very clearly, I suggest eliminating this sentence from the first page so as not to become repetitive.

Authors Response: Line 26 removed.

This article innovates in the sense that studies that approach ageism in the university context are very scarce.

The intro is clear and well-organized.

In the participants section, the first paragraph mainly refers to ethical and procedural issues. That's why I suggest that this section be called "Participants and Procedures"

Authors Response: Added “and Procedures”

When reading the participants' section, I had a concern that I want to share with the authors. Is there not in your study an important confounding factor associated with sampling (particularly with regard to professional status)?

I don't know if we were able to rigorously put 78 instructional faculty and 35 staff members in the same sample. Do you think ageism and negative stereotypes impact these two subpopulations in the same way? Before proceeding with predictive analyses, I would suggest that you perform some simple comparative analyzes between these populations (e.g., t test, using exact test) in relation to the main variables used in the study (mainly ageism and life satisfaction). Only in this way will we be more confident that all these people can be part of the same group for the analyses.

Authors Response: Added a section Preliminary Analyses to show that there were no significant differences on the 8 variables due to employments status

The instruments are very well described and showed excellent reliability indicators.

Table 1 is unformatted, it is not possible to identify the correlational values correctly

Authors Response: Formatted

The regression results do not bring much new information considering what has already been shown in the correlational analyses. But they are well presented, organized and thought out.

Reviewer 2 Report

The subject is of great interest. My recommendations are as follows:

Material and method:

Explain abbreviations (IRB)

Explain adequately the instruments used, especially the items of the scales that have been modified.

Write up the items used in:

·         The Workplace Age Discrimination Scale(WADS) [38]

·         A modified version of the Workplace Intergenerational Climate Scale (WICS) [31]

·         The Negative Aging Meta-Stereotypes scale [39].

Discussion:

Numerical results should not appear.

It is essential that a greater number of existing studies on the subject appear in order to be able to compare and see the difference/similarity with the existing literature.

Conclusions:

These should respond to the objectives. The wording that appears in this section responds more to the limitations of the study than to the conclusions.

Author Response

I found it interesting and enjoyable the way the authors used factor analysis

The discussion adequately discusses the focal results that were found

The subject is of great interest. My recommendations are as follows:

Material and method:

Explain abbreviations (IRB)

Authors Response: Reworded that sentence to clarify IRB.

Explain adequately the instruments used, especially the items of the scales that have been modified. Write up the items used in:  The Workplace Age Discrimination Scale (WADS). A modified version of the Workplace Intergenerational Climate Scale (WICS).

Authors Response:

We have added all of the items that we included in the study and have removed the word modified as the authors do not use the word modified. We used it to indicate that we only used 2 of the 5 subscales.

  • The Negative Aging Meta-Stereotypes scale [39]. (Author’s Reply: Items added)

Discussion:

Numerical results should not appear.

Authors Reply: The frequencies were not reported the Results section and they were reported here to clarify the possibility of the lower variability explanation. The only other numerical value was the rather high r, the only reason we want to keep it is for easy readability and emphasis.

It is essential that a greater number of existing studies on the subject appear in order to be able to compare and see the difference/similarity with the existing literature.

Authors Response: Added more literatures-based comments

Conclusions:

These should respond to the objectives. The wording that appears in this section responds more to the limitations of the study than to the conclusions.

Authors Response: Added sentences to address results pertaining to the objectives of the study.

The article is written in the correct form and I do not have negative comments.

Reviewer 3 Report

The article is written in the correct form and I do not have negative comments.

Dear Authors,

The manuscript addresses important issues related to an aging working population. It is worthwhile to carry out further research in this direction.

I suppose, the age discrimination problem is not just about this place and population.

In my opinion, the manuscript can be published.

Author Response

Dear Authors,

The manuscript addresses important issues related to an aging working population. It is worthwhile to carry out further research in this direction.

I suppose, the age discrimination problem is not just about this place and population.

In my opinion, the manuscript can be published.

Round 2

Reviewer 1 Report

Thanks for the modifications, I'm satisfied with the changes.